# The Antibiotics Degradation and Its Mechanisms during the Livestock Manure Anaerobic Digestion

**DOI:** 10.3390/molecules28104090

**Published:** 2023-05-15

**Authors:** Muhammad Zubair, Zhaojun Li, Rongsheng Zhu, Jiancai Wang, Xinghua Liu, Xiayan Liu

**Affiliations:** 1Institute of Animal Science and Veterinary Medicine, Shandong Academy of Agricultural Sciences, No. 202 Industry North Road, Jinan 250100, China; scientist_zubair@yahoo.com (M.Z.);; 2Beijing Key Lab for Source Control Technology of Water Pollution, Beijing Forestry University, Beijing 100083, China

**Keywords:** anaerobic digestion, livestock, antibiotic resistance gene, antibiotics

## Abstract

Antibiotics are administered to livestock at subtherapeutic levels to promote growth, and their degradation in manure is slow. High antibiotic concentrations can inhibit bacterial activity. Livestock excretes antibiotics via feces and urine, leading to their accumulation in manure. This can result in the propagation of antibiotic-resistant bacteria and antibiotic resistance genes (ARGs). Anaerobic digestion (AD) manure treatment technologies are gaining popularity due to their ability to mitigate organic matter pollution and pathogens, and produce methane-rich biogas as renewable energy. AD is influenced by multiple factors, including temperature, pH, total solids (TS), substrate type, organic loading rate (OLR), hydraulic retention time (HRT), intermediate substrates, and pre-treatments. Temperature plays a critical role, and thermophilic AD has been found to be more effective in reducing ARGs in manure compared to mesophilic AD, as evidenced by numerous studies. This review paper investigates the fundamental principles of process parameters affecting the degradation of ARGs in anaerobic digestion. The management of waste to mitigate antibiotic resistance in microorganisms presents a significant challenge, highlighting the need for effective waste management technologies. As the prevalence of antibiotic resistance continues to rise, urgent implementation of effective treatment strategies is necessary.

## 1. Introduction

The worldwide livestock industry is predicted to increase, and as a result, the use rate of therapeutic antibiotics in the livestock industry is expected to exceed 100,000 tons by 2030 per year [1], although their use as growth promoters has been banned in European and American countries [2]. Livestock, poultry, and fisheries are all fed antibiotics to stimulate growth at subtherapeutic dosages as part of the livestock and fish husbandry production process. Antibiotics are notoriously challenging to degrade. They have bactericidal or antimicrobial actions at high doses, inhibiting bacterial activity or growth. There is a high chance that unmodified or still-active antimicrobials may be detected in environmental samples since animals excrete significant amounts of antibiotics in feces and urine; because antibiotics are poorly absorbed in animal guts, the recovery of antibiotics residues in manure is high [3]. Livestock farms discharge ARG-containing bacteria via drainage, treated wastewater, and solid waste into the environment. Horizontal gene transfer (HGT) can occur between antibiotic-resistant bacteria and indigenous bacteria by conjugation, transformation, and transduction. Antibiotic-resistant bacteria have the potential to enter human bodies through various means, including ingestion of contaminated food or water, as well as through occupational exposure such as inhalation [4]. The amount of veterinary antibiotics used in animal feed as a disease preventive agent has grown significantly during the last two decades. For example, veterinary antibiotic consumption in China increased by 6000 tons annually [5]. However, livestock has a limited ability to metabolize veterinary antibiotics, typically only converting 10% to 30% of the antibiotics consumed. Consequently, a considerable amount of antibiotics is excreted in the feces as either metabolites or in their original form, often at concentrations reaching several hundred micrograms per liter [6].

The use of antibiotics in livestock production is primarily non-therapeutic, serving to promote growth and prevent disease [7]. These antibiotics are typically administered at low, subtherapeutic levels in the animals’ gastrointestinal tracts, which inhibits the development of susceptible bacterial populations [8]. However, this practice can exert selective pressure on microorganisms in the gut to acquire and maintain antibiotic resistance genes (ARGs), leading to an increase in the prevalence of resistant bacterial populations [9]. The spread of ARGs to adjacent ecosystems and their subsequent emergence as pollutants can lead to environmental pollution due to antibiotic-resistant bacteria (ARB) being excreted by animals into receptive environments such as soil and water. Further ARG replication and dissemination increase the risk of human exposure, especially for agricultural workers and nearby residents [4]. The associated ARGs in the soil–plant system highlights a possible pathway for the transmission of ARGs into the human microbiome and pathogens via the food chain [10]. While most antibiotics are stable during manure storage, anaerobic digestion can break down and remove them to variable degrees depending on the antibiotic concentration and class, bioreactor operating conditions, feedstock type, and inoculum source. The following sequence is frequently used for antibiotic degradation: anaerobic digestion (AD), manure storage, composting, and soil [6]. AD has been tested for the removal of antibiotics and ARGs from manure, and it is one of the approaches to organic waste management in general. It is thought to be a win-win situation for both energy generation and environmental risk reduction [11]. Manure accounts for over 70% of total biogas input [12], and AD of manure removes dangerous pollutants such as antibiotics and ARGs. The issue with antibiotics in manure is not just their impact on the environment. Studies have shown that antibiotics can decrease methanogenesis of anaerobic bacteria, resulting in reduced methane or biogas generation [13].

## 2. Antibiotics: Types and Concentrations in Livestock Manure

Antibiotics are not just used in clinical settings to treat diseases in people. Antibiotics are also used in the livestock husbandry, administered in subtherapeutic doses in concentrated animal feed to promote feed conversion efficiency, growth and disease prevention [14]. Antibiotics used in agriculture and livestock medicine are closely connected (belong to the same general classes, perform similar functions, and act in similar ways) to those antibiotics used in human medicine [15]. Due to differences in food animal species, geographical production patterns, kinds of production systems, intense or extended farming, and the purpose of farming (commercial, industrial, or home), as well as population size and socioeconomic position, antibiotic usage and consumption patterns demonstrate significant geographic variety between continents [1]. Animal feed containing non-essential antibiotics to promote growth remains unregulated in developing nations. The continued use of non-essential antibiotics (use for other purposes such as growth promotion or disease prevention) in livestock farming is due to increased farmland concentration, insufficient government strategies and controls, concerns about the sale and use of antibiotics, the lack of infection control measures, and farmers’ resistance to implementing mandated changes to farming methods. Antimicrobial growth promoters are still used in developing countries to keep livestock healthy, boost output, and increase farmer income [16]. However, European agricultural data show no loss of productivity due to the restriction [17].

Livestock are treated therapeutically and subtherapeutically with many antibiotic classes, [18], including:(1)Polyether antibiotic: monensin, narasin, lasalocid, and salinomycin;(2)Fluorochinolones: enrofloxacin, ofloxacin, norfloxacin, danofloxacin;(3)Sulphonamides: sulphadimidine, sulphamethoxazole, trimethoprim, sulphadoxine;(4)Macrolides: azithromycin, clarithromycin, clindamycin, erythromycin, roxithromycin, spiramycin, tylosin, and vancomycin;(5)Tetracyclines: chlortetracycline, doxycycline, oxytetracycline, and tetracycline;(6)b-Lactams, penicillins: amoxicillin, ampicillin, oxacillin, piperacillin, benzylpenicillin, cloxacilin, dicloxacilin, flucloxacillin, methicillin, phenoxymethylcillin, mezlocillin, and nafcillin.

In Shandong, China, 126 pig manure samples taken from 21 animal-feeding operations showed tetracyclines (85–97%), sulphonamides (52%), and macrolides (5%) [19]; the same results were found in other parts of China [20] and Japan [21]. Antibiotics are excreted directly to urine and feces (17–90% for livestock) [22], unaltered or as living by-products of the parent species [23].

Table 1 presents the excretion rates of antibiotics by animals, expressed as the concentration of antibiotic residues in animal manure. It is important to note that these concentrations may differ from the original dosage due to metabolic processes within the animal. Additionally, some antibiotic metabolites may be more toxic than their parent compounds, and certain metabolites, such as acetic-conjugated sulphonamides, may revert to their original molecules over time. Animal manure is a significant source of antibiotic residues in the environment. It has been shown to contribute to the spread of antibiotic-resistant bacteria, particularly when non-metabolized drug residues are present [24].

The concentrations of antibiotics in manure usually range from 1 to 10 mg·kg^−1^ or L^−1^ but can exceed 200 mg·kg^−1^ or L^−1^ [32], while in China, it is 100 mg·kg^−1^ or L^−1^ [33]. The accuracy and reliability of the reported amounts of antibiotic excreted by animals may be influenced by variations in the extraction and quantification procedures used in these investigations. It is important to note that suboptimal extraction and quantification methods can result in a wide range of observed values and may not accurately reflect the true levels of antibiotic excretion. Therefore, further optimization and standardization of these procedures is necessary to improve the precision and comparability of results across studies. Antibiotics have been found in swine, cattle, and poultry/turkey dung with concentrations of up to 216 mg·L^−1^ [22]. As indicated in Table 2, the measured quantities of antibiotics in diverse environmental samples give solid evidence of their ubiquity in manure.

The measurement of antibiotics in complex matrices such as biological sludge and soil is not standardized, making comparisons between studies difficult. The majority of research does not even describe the manure storage and management practices prior to sampling when reporting results. The liquid/solid antibiotic partitioning affects the outcomes. In contrast to overall concentration, sampling leaching manure reveals the percentage of antibiotics in the solid phase. During storage, the majority of antibiotic residues in manure combine to form stable compounds with soluble organics. Some antibiotics become mobile when manure is applied to fields and contaminate the soil and groundwater in the area. The features of the antibiotic, the soil, and the hydrological effects all play a role in this. Understanding the rates of biodegradation and the potency of degraded antibiotics in various situations requires more investigation (soils, manures, and wastewater).

ARGs give resistance to nine significant families of antibiotics, including fluoroquinolone, chloramphenicol, and amphenicol (FCA), tetracyclines (tet), sulfonamides (sul), β-lactams (bla), and macrolide-lincosamide-streptogramin B (MLSB). The ARG classes that are most frequently found in cattle manure are tet, sul, erm, fca, and bla [4]. Van Boeckel et al. [1] conducted an analysis of antimicrobial consumption in animal production systems, and observed significant differences in average usage rates across species. Specifically, the mean amount of antimicrobial agents consumed per kilogram of animal produced was estimated to be 45 mg/kg for cattle, 148 mg/kg for chickens, and 172 mg/kg for pigs [35]. Furthermore, antimicrobial dosage pattern dependent on animal life stage. Swine are given higher antibiotic dosages early in life and lower amounts later. Finisher swine waste has less ARG abundance and diversity than sow and nursery waste. Chickens are raised using antibiotics throughout their life cycle, therefore their feces may contain more ARG than other livestock feces after two months of hatching to slaughter. Cattle waste had less ARGs than poultry or swine waste, possibly due to delayed exposure to antibiotic feed additives. Antibiotics are provided to meat and dairy cattle to treat and prevent disease (primarily to avoid mastitis, an inflammation of the mammary gland and udder tissue, and liver abscesses). They are not used therapeutically in feedlots [36]. ARG abundance in livestock feces varies by country. ARGs varies widely across countries and regions, with some areas experiencing higher levels of ARGs than others. According to a recent meta-analysis of 96 countries, China was found to have the highest absolute abundance of ARGs, likely due to its status as the world’s largest producer and consumer of antibiotics [37]. Despite similar levels of antibiotic consumption on a global scale (with both Shandong and Colorado accounting for 23% and 19% of worldwide use in 2010, respectively), there is no correlation between antibiotic consumption and the abundance of antibiotic resistance genes (ARGs). This is exemplified by the fact that Shandong’s ARG abundance is 10 times higher than that of Colorado, contradicting the notion that higher antibiotic consumption necessarily leads to increased ARG prevalence [38]. The abundance of antibiotic resistance genes (ARGs) can vary significantly between countries, influenced by country-specific factors. For instance, the quantity of ARGs present in cattle dung from Shandong is approximately 100 times greater than that found in Shaanxi (around 1011 copies/g) [39]. ARG abundance can vary significantly between dairy farms in different regions, as seen in the differences between midwest and northeastern farms. Although there may be some correlation between ARG abundance and antibiotic use or residual antibiotics in certain countries, the proliferation of ARBs and the dynamics of ARG propagation and attenuation are influenced by site-specific physical and chemical factors, leading to considerable variability [4].

## 3. The Fate of Antibiotics and Antibiotics Resistance Genes during Anaerobic Digestion

### 3.1. Potential of Anaerobic Digestion to Degrade Antibiotics in Livestock Manure

Globally, the livestock business utilizes around 60,000 tons of antibiotics per year, such for basic veterinarian care, prophylaxis, and growth promotion is a significant contributor of ARGs to the environment [40]. Various waste treatment processes are being researched to minimize ARGs in livestock manure before land application. Depending on the class and concentration of antibiotics, the operating conditions of the bioreactor, the kind of feedstock, and the source of the inoculum utilized, AD has the capacity to break down antibiotics in manure to varying degrees. Additionally, AD can minimize the concentrations of antibiotics often found in manures while causing minimal disruption to the process’s stability and output [41]. Recent research has revealed that AD could be a viable option for lowering ARG levels in various animal manures [42], albeit total elimination is unlikely. The effectiveness of ARG removal appears to be very dependent on a number of parameters. According to research, pre-treatment, temperature, residence time, and post-treatment are the key parameters determining ARG elimination during anaerobic digestion of livestock manure [43] (see Table 3 below). In general, thermophilic environments have proven to remove more ARG than mesophilic environments [44]. Longer residence times are associated with higher levels of ARG removal [45], while slow mixing rates and low-solids-content manures have been proven to remove less ARG [46]. The majority of pre-treatment research has focused on thermal pre-treatment, which involves high temperatures (60–180 °C) for a short residence time (less than 80 min), and post-treatment methods such as ammonia removal, additional storage, composting, and thermophilic aerobic digestion, among others [43]. Furthermore, residual antibiotics and other contaminants such as heavy metals, biochar, or lignite can reduce the efficacy of ARG removal [41].

### 3.2. Anaerobic Digestion

If pathogens are not effectively inactivated prior to the use of manure in crop cultivation or discharge into the environment, there is a risk of foodborne illness due to the potential contamination of crops or water sources. Common foodborne pathogens, such as *Salmonella*, *E. coli*, and *Listeria*, can be present in biowaste, and proper handling and treatment of this waste is crucial to prevent the spread of pathogens and protect public health. Pathogens present in food, water, or air can pose a risk to human health when proper waste management and treatment practices are not followed. Factors such as the increase in food waste, animal manure, and other biowaste, combined with inadequate pathogen inactivation during waste management, can contribute to the spread of biosafety-related diseases [47]. Livestock wastes are a major source of pollution because they include significant levels of organic compounds, nitrogen, phosphorus, and pathogens. For example, animal wastes contribute 55% COD, 22% TN, and 32% TP to water pollution in China [48]. Manure treatment processes such as AD have become increasingly popular due to their ability to reduce organic matter pollution and pathogens and produce methane-rich biogas, a renewable energy source. While numerous parameters influence AD processes, digestion temperature, pH, total solids concentration, substrate type, organic loading rate (OLR), hydraulic retention time (HRT), intermediate substrates, and pre-treatments are known to be the most important [49].

To achieve effective pathogen inactivation while producing bio-methane, it is crucial to comprehend and measure the impact of different operational conditions on the inactivation of pathogens. Ma et al. [47] performed a meta-analysis on various substrates and found a statistically significant impact of AD on pathogen inactivation (*p* < 0.01). The study included 63 investigations of fecal coliform inactivation, 51 investigations of *E. coli* inactivation, and 38 investigations of salmonella inactivation. More data were available for mesophilic AD processes than for either ambient or thermophilic AD processes, indicating a preference for mesophilic systems. Understanding and quantifying the effect of different parameters on pathogen inactivation is critical for balancing the operating conditions of pathogen inactivation and bio-methane production.

#### 3.2.1. Operating Temperature

The microbiome is largely shaped by the temperature, but the microbial populations in the digester may have an impact on the proliferation of ARGs [43]. Studies from the past suggested that ARGs could behave differently to operational temperatures. Numerous types of research have been carried out to examine the dynamics of ARGs and bacterial communities under various anaerobic digesting conditions. In AD, mesophilic conditions are those that promote the growth of microorganisms at temperatures between 25 and 40 °C. This range of temperature is the most common in AD and is ideal for the growth of a wide range of microorganisms. This temperature range is also optimal for the digestion of most organic materials. Thermophilic conditions, on the other hand, promote the growth of microorganisms at temperatures between 45 and 65 °C. Thermophilic conditions are more effective than mesophilic conditions in breaking down complex organic compounds such as proteins, lipids, and lignocellulose, and are ideal for treating certain types of waste, such as sewage sludge or food waste. Moderate conditions are those that fall between mesophilic and thermophilic conditions, typically between 40 and 45 °C. The prevalence of cefazolin-resistant bacteria and the genes ermB, aphA2, and bla may be significantly decreased by mesophilic anaerobic digestion [50]. Temperature is a crucial factor in anaerobic digestion, which produces biogas, ARGs, and the succession of microbial communities. Consequently, temperature changes can have a variety of consequences on these processes [51]. Diehle et al. [51] suggested that mesophilic anaerobic digestion did not reduce ARGs as effectively as thermophilic anaerobic digestion. Thermophilic digestion, according to Ma et al. [52], was superior to mesophilic digestion in reducing the ermB, ermF, tetO, and tetW genes, but it performed similarly to or less effectively at reducing other ARGs and intI1. According to Sun et al. [44], ARGs and integrase genes are impacted by the succession of bacterial communities. More than moderate and mesophilic treatments, thermophilic anaerobic digestion decreased mesophilic ARG-carrying bacteria. Because anaerobic digestion eliminated aerobic integron hosts, it reduced overall integrase gene abundance. Thermophilic digestion is faster, more stable, and produces more methane than mild and mesophilic digestion. However, a few studies revealed the reverse pattern, notably insufficient ARG elimination in thermophilic AD. For instance, Huang et al. [53] found that the ARG abundance of AD pig manure was greatest at 55 °C compared to 25 °C and 37 °C. Additionally, Sun et al. [54] observed that the thermophilic solid-state AD of cattle manure had a 23.7% greater ARG abundance than the mesophilic AD and Proteobacteria, a possible host for an ARG, were found to be more prevalent in thermophilic AD. This research hypothesized that these observations may be explained by the increased ARB activity in thermophilic environments. In addition, studies showed that ARG subtypes respond differently depending on the temperature. The fate of ARGs in waste-activated sludge’s thermophilic and mesophilic AD was compared by Zhang et al. [55]. Different ARG subtypes (tetG, tetO, tetW, and ermB) were successfully eliminated by mesophilic AD, but sul2 was successfully eliminated by thermophilic AD. However, the elimination of sul1 was unsuccessful at any temperature. The study conducted by Flores-Orozco et al. [56] utilized meta-analysis to demonstrate that thermophilic anaerobic digestion (AD) is a more efficient means of reducing antibiotic resistance genes (ARGs) in pig manure when compared to cattle manure. This finding implies that the type of feedstock employed has a significant impact on determining the most suitable operating temperature for the effective removal of ARGs. Low ARGs elimination in cow manure may account for excessive free ammonia levels [54]. Unlike other process factors, the effects of temperature on ARG removal have been extensively studied. The ideal temperature for removing ARGs and mobile genetic elements (MGEs) during AD cannot be determined, nevertheless, and no definitive conclusions can be drawn from this. Additionally, variations in initial ARG/MGE levels and subsequent removal efficiencies may be caused by variations in other process factors, such as feedstock type, solid content, and residence periods.

#### 3.2.2. Two-Stage Anaerobic Digestion

In AD, acidogenic bacteria and methanogenic bacteria interact syntrophically. However, their growth kinetics, nutrition, and optimal environment factors have distinctions between them. Researchers have proposed the acidogenic phase followed by the methanogenic phase (two-stage AD) to give the best growth conditions for each microbial community [57]. However, few researchers examined the impact of such a two-stage design on ARG elimination. Wu et al. [58] investigated how two-stage digesters eliminated ARG and MGE. Thermophilic AD was more effective at removing ARG than mesophilic AD. However, during thermophilic AD in the second, 80% of the first-stage ARGs returned. The variety of microorganisms and the number of possible ARBs were linked with ARG profiles. The findings showed that ARG growth could be hampered by a less varied microbiome in a thermophilic–acidogenic environment. Shi et al. [57] recently proposed that methanogenesis and acidogenic stages would have differing effects on ARG propagation. They found that the initial stage of alkaline fermentation in thermophilic and mesophilic environments elevated the genes for resistance to macrolide, lincosamide, and streptogramin. In contrast, under thermophilic and mesophilic environments, the second stage reduced these genes. As a result, the process conditions may influence the composition of the microbial community and ARGs fate.

#### 3.2.3. Residence Time

The performance of ADs and the digester microbiota depend on solids residence time (SRT), according to numerous research studies. Only a small number of studies, meanwhile, specifically looked into how SRTs affected how ARGs turned out. Ma et al. [52] looked at the removal of ARG during the period of 10–20 days of SRTs while primary and secondary sludge were co-digested in a mesophilic environment. Their research suggests that a 20-day SRT in mesophilic AD could be able to eliminate more distinct ARG subtypes than shorter SRTs, including tetC, tetG, tetX, sulI, and suII. Sun et al. [59] hypothesized that increased ARG elimination could be linked to the limited microbial diversity caused by the oligotrophic environment generated under prolonged residence durations. As a result, some ARB would be removed, and the likelihood of new cells proliferating might be decreased. Additionally, he stated that depending on the operational temperatures, residence times can have insignificant influence on ARG elimination. In conclusion, a variety of factors may affect the best SRTs for removing ARGs. In order to achieve effective ARG removal during AD, a systematic adjustment of SRTs taking other parameters into account would be required.

#### 3.2.4. Total Solids

The AD with total solids ≥ 15% is called solid or dry AD. In addition to having better volumetric methane production than wet AD, residues of dry AD are frequently applied to agriculture land without dewatering procedures [60]. According to a few recent research, dry AD digestate should contain less ARG than wet AD [61]. The ARG removal in dry and wet mesophilic AD of cattle manure was compared by [54]. The dry AD was superior to a wet AD in ARG removal. The dry AD saw a drop in ARG and ARB abundances of at least seven genes. Consistent with these findings, another study found that dry AD was superior to wet AD in eliminating ARG and MGE during the co-digestion of food waste and pig manure [61]. A recent study found that microbial mobility is essential for the HGT of ARGs. It is activated mixing is rarely used in dry AD systems, which may reduce their susceptibility to ARG spread [62]. Contradictory results were reported by Sui et al. [63], who found that dry AD of swine manure was less effective at removing ARG than wet AD. The authors hypothesized that the accumulation of free ammonia nitrogen and volatile fatty acids due to elevated TS levels was responsible for the poor ARG removal in dry AD. As a result, we need a further systematic study to fully comprehend the impact of control factors on ARG propagation in the dry AD process.

#### 3.2.5. Co-Digestion

For AD to function properly, the carbon-to-nitrogen ratio, toxicity, pH, moisture, and trace element levels must all be optimized [64]. Co-digestion of two or more complementary feedstocks is frequently regarded as a suitable strategy to accomplish these goals. Few researchers have looked into what happens to ARGs during co-digestion so far [65,66]. Co-digesting with a variety of different feedstocks could affect microbial community development, which in turn could affect ARG abundances and quality. In addition, the selection pressure for ARG propagation may be mitigated by diluting antibiotics through feedstock mixing [67]. Song et al. [68] suggested that ARG removal can be aided by co-digesting wheat straw and swine manure at the optimal mixing ratio. According to [66], combining swine manure and wheat straw with Chinese medicinal herbal residues as a co-substrate can reduce HGT by means of microbial gluconate oxidase. However, it was recognized that minimizing the spread of ARGs would require careful consideration of the co-substrate selected and their optimal mixing ratios [67].

## 4. Required Future Research

Since the topic of antibiotic use and ARGs are still relatively new, there is still a need to establish a comprehensive body of existing knowledge corresponding to diverse manure treatment procedures and feedstock characteristics. Future research and current understanding are anticipated to contribute to the development of better treatment techniques that can potentially lessen the load of ARGs that waste materials could potentially release into the environment. It is critical to develop standardized techniques for assessing the impact, degradation, and fate of various antibiotics and their metabolites during anaerobic digestion in order to reliably, compare data from different studies. It is necessary to concentrate on kinetic and metabolic modeling and simulation of inhibition, recovery, and adaptive processes in order to evaluate the impact of culture matrices, solid content, and the composition of the organic manure fraction on the de-radiation dynamic of various antibiotics during anaerobic digestion. Research is particularly essential to ascertain how to most accurately predict the relative relevance of physical and chemical vs. living organism breakdown processes for different types of antibiotics. It is important to understand and quantify the impacts of manure fractioning on the anaerobic breakdown of antibiotics in different manure classes, antibiotics kinds, reactor designs, and operations. For design purposes, it is vital to investigate how various anaerobic reactor designs and operations affect the kinetic data for antibiotic parent chemicals and their metabolites. To avoid problems with operations caused by the harmful effects of high concentrations of antibiotics on anaerobic digestion, it is important to look into how to process staging and adjustment affect things. The possibility of psychrophilic anaerobic digestion of different livestock manures for the removal of antibiotics and antibiotic-resistant microbes has not been studied.

Antibiotic resistance in microorganisms is a difficult problem that necessitates advancements in livestock waste treatment procedures. Antibiotic resistance among infections is increasing the demand for efficient treatment methods. The following are some of the knowledge gaps and research needs for the future:The influence of operational parameters (pH, free available chlorine, HRT, SRT, and biomass content) and environmental variables (temperature, COD, BOD, and water flow) on the production of ARB and ARGs during wastewater treatment.The identification of novel mechanisms of ARG development as well as the determination of the dominant processes of ARG development (mutation, selection, and genetic exchange techniques such as conjugation, transduction, and transformation).Future research should focus on the wider spectrum of ARBs and ARGs resistance, including fluoroquinolone, ertapenem, and levofloxacin resistance.

## 5. Conclusions

In conclusion, it was evident that optimizing various process parameters could successfully eliminate ARGs during AD. Previous research has identified certain fundamental factors that influence the proliferation of ARGs in AD. However, the interactions between different process factors often led to contradictions in prior findings. Therefore, further research is required to fully comprehend how the interplay between various process factors impacts the proliferation of ARGs in AD. Additionally, continued development of conductive additives and pre-treatment procedures is crucial to provide dual solutions for better biogas generation and ARG management, ensuring economic sustainability of AD while also decreasing the danger of digestate for land application.

## Figures and Tables

**Table 1 molecules-28-04090-t001:** Comparative excretion rates of commonly used animal antibiotics.

Antibiotics	Manure Source	Excretion Levels (%)	Status	Reference
*Tetracycline*	Animal feces	25%	Not reported	[25]
*Chlortetracycli*	Steers feces	75%	Not reported	[26]
*Oxytetracycline*	Calves manure (feces, urine, and bedding)	23%	Unchanged	[27]
*Oxytetracycline*	Castrate sheeps	21%	Unchanged	[28]
*Chlortetracycline*	Young bulls	17–75%	Unchanged	[28]
*Tylosin*	Pigs	40%	Unaltered or as potent metabolites	[25]
*Monensin*	Beef cattle feces	40%	Unchanged	[29]
*Virginiamycin*	Piggeries liquid manure	20%	After several days of storage	[30]
*Tylosin*	Urine	50–60%	Not reported	[31]

**Table 2 molecules-28-04090-t002:** The concentration of antibiotics present in manures.

Antibiotic	Matrix	Concentration	Reference
*Oxytetracycline*	Manure	136 mg·L^−1^	[14]
*Chlortetracycline*		46 mg·L^−1^	
*Tetracycline*	Swine manure	98 mg·L^−1^	[11]
*Oxytetracycline*		354 mg·L^−1^	
*Chlortetracycline*		139 mg·L^−1^	
*Doxycycline*		37 mg·L^−1^	
*Sulfadiazine*		7.1 mg·L^−1^	
*Tetracycline*	Swine manure	30 mg·kg^−1^ DM	[18]
*Sulphonamides*		2 mg·kg^−1^ DM	
*Tylosin*	Fresh calf manure	0.11 mg·kg^−1^	[34]
*Oxytetracycline*		10 mg·kg^−1^	
*Enrofloxacin*	Chicken manure	1420.76 mg·kg^−1^	[34]
*Chlortetracycline*	Beef manure stockpile	6.6 mg·kg^−1^	[28,35]
*Monensin*		120 mg·kg^−1^	
*Tylosin*		8.1 mg·kg^−1^	
*Chlortetracycline*	Swine manure	764.4 mg·L^−1^	[12]
*Chlortetracycline*	Swine manure storage lagoon	1 mg·L^−1^	[36]
*Oxytetracycline*		0.41 mg·L^−1^	
*Oxytetracycline*	Cow manure	0.5–200 mg·L^−1^	[37]

**Table 3 molecules-28-04090-t003:** The effect of anaerobic digestion on antibiotic residues in manure: reduction after in concentrations anaerobic digestion.

Treatments	Antibiotics	Concentrations	Observed Reductions
Anaerobic digestion of cattle manure, 28 days	*Monensin*	10.74 mg·L^−1^	3% (22 °C)
		10.36 mg·L^−1^	8% (38 °C)
		10.30 mg·L^−1^	27% (55 °C)
Anaerobic digestion of swine manure, 21 days	*Chlortetracycline*	6.5 mg·L^−1^	7% (22 °C)
		8.3 mg·L^−1^	80% (38 °C)
		5.9 mg·L^−1^	98% (55 °C)
Swine manure from lagoons	*Tylosin*	10–400 mg·kg^−1^	95–75%
Anaerobic sequence batch reactor (ASBR)	*Tylosin A*	1.6 mg·kg^−1^	Degraded to <detection limit
		5.8 mg·kg^−1^	Decreased to 0.01 mg·L^−1^ in 48 h
Batch anaerobic digestion	*Oxytetracycline*	20 mg·L^−1^	55–73% at 7 °C
Thermophilic anaerobic digestion	*Clarithromycin*	557 μg·L^−1^	36% in 40 days
Thermophilic anaerobic digestion	*Erythromycin*	356 μg·L^−1^	90% in 40 days

## Data Availability

Not applicable.

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
