# Peer review of "The Antibiotics Degradation and Its Mechanisms during the Livestock Manure Anaerobic Digestion"

_molecules, 2023, doi:10.3390/molecules28104090_

Round 1

Reviewer 1 Report

The manuscript is not easy and fluent to read, the different paragraphs look like having been written by different authors without overall uniformity, some are chaotic in the expression of concepts, often with useless repetitions of sentences and concepts, or sentences out of context, making difficult for the reader to perceive the whole. The English language is sometimes poor and difficult to interpret, proofreading by a native is recommended.

The references are up to date and the whole section is complete and abundant, revealing the meticulous preparatory work that the authors have performed.

16-18 and 41: not all antibiotics have poor absorption from the oral route, otherwise we would not use it; please, specify

20 and further: please, specify acronyms at first use

35-37: in many countries the growth-promoting use of antibiotics has not been allowed for years, please, specify

40: if the authors refer to the administration of sub-therapeutic doses, the eliminations cannot be "huge"

45-47: sentence difficult to understand, please rephrase

49-52: sentence difficult to understand, please rephrase

53: drugs are always administered at sub-lethal doses to animals, regardless they have a therapeutical or growth-promotion goal, unless they’re euthanasia drugs

53-57: paragraph difficult to understand, please rephrase

69: “…and soil” what? Please, check the sentence as it seems something is missing

90: what is the meaning of “non-essential antibiotics”? please, explain

95: reference’s title is “Restricting antimicrobial use in food animals: lessons from Europe”, so not only Sweden

100: ciprofloxacin is rarely used in veterinary medicine (pet only), while enrofloxacin is the commonest veterinary fluoroquinolone, whose metabolism produces ciprofloxacin, reasons for the finding of resistance also from veterinary use

102: “trimethoprim, sulphadoxine” should continue previous line

108: "italic" is already indicative of a writing style, it is not necessary to specify, and at the same time the whole list 99-107 is in italics, so it is not clear which other drugs we should refer to

114: the caption of table 1 reports ""animal antibiotic excretion rate" (which incidentally is also not very descriptive of the content of the table itself), not the concentrations

114-119: chaotic description of the fate of parent molecules and metabolites, please rephrase

125: what do authors mean with “insufficient extraction and quantification procedures”? recovery % is used to correct results in field samples, so that they can be compared (see also 130-131)

147-151: sentence difficult to understand, please rephrase; furthermore, “macrolide-lincosamidreptogramin B” is repeated twice, “-lactams” needs a “beta”, and check lincosamidreptogramin”

152-154: the classes of antibiotics used in the different species vary, what does the cumulative "average antimicrobial agent consumed per kilogram of animal produced" mean? and please, check for sentence construction

157-158: sentence difficult to understand, please rephrase

160-161: please, explain the sentences (and the meaning of “liver abscesses”, as the main bacterial disease of dairy caws is mastitis)

162-173: paragraph difficult to understand, please rephrase

176: what is the meaning of “essential pathogens”? please, explain

177-179: sentence difficult to understand, please rephrase

190-198: paragraph difficult to understand, please rephrase

204-205: please explain “moderate, mesophilic ant thermophilic” conditions

232-233: quite obvious, as microbiomas of different species are different

Table 3: please, make caption more descriptive of table content

316: 3.2 section is more explanatory of the procedure description, so maybe it could be convenient to move it as 3.1, to improve the comprehension of the text

353-354: this is a scientific paper, maybe the reader already has some knowledge about biotic and abiotic

361-363: sentence out of context

379-388: paragraph difficult to understand, please rephrase

Author Response

Response to Reviewer 1 Comments

The manuscript is not easy and fluent to read, the different paragraphs look like having been written by different authors without overall uniformity, some are chaotic in the expression of concepts, often with useless repetitions of sentences and concepts, or sentences out of context, making difficult for the reader to perceive the whole. The English language is sometimes poor and difficult to interpret, proofreading by a native is recommended.

The references are up to date and the whole section is complete and abundant, revealing the meticulous preparatory work that the authors have performed.

Response: Thank you for your feedback on the manuscript. Our team appreciates your comments and we are sorry to hear that the paper was difficult to read. There are a few areas in which the manuscript can be improved, including coherence, uniformity, and clarity. As part of the revision process, we will pay particular attention to the organization and structure of the paper, as well as the language and style of writing. Additionally, we will ensure that all sentences are presented in their proper context and eliminate any unnecessary repetition of sentences and concepts.

I am pleased to hear that you found the preparatory work meticulous and appreciate the positive comments you made regarding the references section. The references will continue to be updated and complete. It is acknowledged that the English language could use some improvement, so we will have a native speaker proofread the manuscript before submitting it. I would like to thank you once again for your valuable feedback. The manuscript will be improved based on your comments.

Point 1: 16-18 and 41: not all antibiotics have poor absorption from the oral route, otherwise we would not use it; please, specify

Response 1: Thank you for pointing out this lapse. The abstract has been revised “Antibiotics are administered to livestock at sub-therapeutic levels to promote growth, and their degradation in manure is slow. High antibiotic concentrations can inhibit bacterial activity. Livestock excrete antibiotics via feces and urine, leading to their accumulation in manure. This can result in the propagation of antibiotic-resistant bacteria and antibiotic resistance genes (ARGs). Anaerobic digestion (AD) manure treatment technologies are gaining popularity due to their ability to mitigate organic matter pollution, pathogens, and produce methane-rich biogas as a renewable energy. AD is influenced by multiple factors, including temperature, pH, total solids (TS), substrate type, organic loading rate (OLR), hydraulic retention time (HRT), intermediate substrates, and pre-treatments. Temperature plays critical role, and thermophilic AD has been found to be more effective in reducing ARGs in manure compared to mesophilic AD, as evidenced by numerous studies. This review paper investigates the fundamental principles of process parameters affecting the degradation of ARGs in anaerobic digestion. The management of waste to mitigate antibiotic resistance in microorganisms presents a significant challenge, highlighting the need for effective waste management technologies. As the prevalence of antibiotic resistance continues to rise, urgent implementation of effective treatment strategies is necessary.” (page 1; Line 15-29).

Point 2: 20 and further: please, specify acronyms at first use

Response 2: Thank you for pointing out this lapse. The Anaerobic Digestion (AD) manure treatment technologies ... has been added and others acronyms also added in revised version

Point 3: 35-37: in many countries the growth-promoting use of antibiotics has not been allowed for years, please, specify

Response 3: Thank you for your suggestion. The reference has been added to the “although their use as grow promoters has been banned in European and American countries [2]” (page 1; Line 35-36).

Point 4: 40: if the authors refer to the administration of sub-therapeutic doses, the eliminations cannot be "huge"

Response 4: Thank you for your suggestion. The word “huge” has been replaced with the “significant amount” of revised version (page 1; Line 41).

Point 5: 45-47: sentence difficult to understand, please rephrase

Response 5: Thanks for your valuable suggestion. The sentence has been rephrased “Antibiotic-resistant bacteria have the potential to enter human bodies through various means, including ingestion of contaminated food or water, as well as through occupational exposure such as inhalation.” (Page 2; Line 46-48).

Point 6: 49-52: sentence difficult to understand, please rephrase

Response 6: Thanks for your valuable suggestion. The sentence has been rephrased “However, livestock has a limited ability to metabolize veterinary antibiotics, typically only converting 10% to 30% of the antibiotics consumed. Consequently, a considerable amount of antibiotics is excreted in the feces as either metabolites or in their original form, often at concentrations reaching several hundred micrograms per liter” in revised version (page 2; Line 51-55).

Point 7: 53: drugs are always administered at sub-lethal doses to animals, regardless they have a therapeutical or growth-promotion goal, unless they’re euthanasia drugs, 53-57: paragraph difficult to understand, please rephrase

Response 7: Thank you for your constructive suggestion. The sentence has been rephrased “The use of antibiotics in livestock production is primarily non-therapeutic, serving to promote growth and prevent disease [7]. These antibiotics are typically administered at low, sub-lethal levels in the animals' gastrointestinal tracts, which inhibits the development of susceptible bacterial populations [8]. However, this practice can exert selective pressure on microorganisms in the gut to acquire and maintain antibiotic resistance genes (ARGs), leading to an increase in the prevalence of resistant bacterial populations.” in revised version (page 2; Line 56-59).

Point 8: 69: “…and soil” what? Please, check the sentence as it seems something is missing

Response 8: Thank you for pointing out this lapse. The sentence has been rephrased “The spread of ARGs to adjacent ecosystems and their subsequent emergence as pollutants can lead to environmental pollution due to antibiotic-resistant bacteria (ARB) being excreted by animals into receptive environments such as soil and water.” in revised version (page 2; Line 59-64).

Point 9: 90: what is the meaning of “non-essential antibiotics”? please, explain

Response 9: Thank you for your constructive suggestion. Non-essential antibiotics are antibiotics that are not used for treating sick animals but are used for other purposes such as growth promotion or disease prevention. The use of non-essential antibiotics in livestock farming has been linked to the development of antibiotic-resistant bacteria. In sentence “(use for other purposes such as growth promotion or disease prevention)” was added in revised version (page 2; Line 93-94).

Point 10: 95: reference’s title is “Restricting antimicrobial use in food animals: lessons from Europe”, so not only Sweden

Response 10: Thank you for pointing out this lapse. The “Sweden” was replaced by “European” in revised version (page 2; Line 99).

Point 11: 100: ciprofloxacin is rarely used in veterinary medicine (pet only), while enrofloxacin is the commonest veterinary fluoroquinolone, whose metabolism produces ciprofloxacin, reasons for the finding of resistance also from veterinary use

Response 11: Thank you for sharing that information with me. It’s true that ciprofloxacin is rarely used in veterinary medicine and enrofloxacin is the commonest veterinary fluoroquinolone. The metabolism of enrofloxacin produces ciprofloxacin. The “ciprofloxacin” has been replaced with “enrofloxacin” of revised version (page 3; Line 104).

Point 12: 102: “trimethoprim, sulphadoxine” should continue previous line

Response 12: Thank you for pointing out this mistake. We have corrected this (page 3; Line 105).

Point 13: 108: "italic" is already indicative of a writing style, it is not necessary to specify, and at the same time the whole list 99-107 is in italics, so it is not clear which other drugs we should refer to

Response 13: Thank you for pointing out this lapse. The sentence has been deleted.

Point 14: 114: the caption of table 1 reports ""animal antibiotic excretion rate" (which incidentally is also not very descriptive of the content of the table itself), not the concentrations

Response 14: Thank you for pointing out this lapse. We have corrected this issue following your comments as “Table 1. Comparative Excretion Rates of Commonly Used Animal Antibiotics” of revised version (page 3; Line 126).

Point 14: 114-119: chaotic description of the fate of parent molecules and metabolites, please rephrase

Response 14: Thank you for pointing out this lapse. The whole paragraph has been rephrased “Table 1 presents the excretion rates of antibiotics by animals, expressed as the concentration of antibiotic residues in animal manure. It is important to note that these concentrations may differ from the original dosage due to metabolic processes within the animal. Additionally, some antibiotic metabolites may be more toxic than their parent compounds, and certain metabolites, such as acetic-conjugated sulphonamides, may revert to their original molecules over time. Animal manure is a significant source of antibiotic residues in the environment. It has been shown to contribute to the spread of antibiotic-resistant bacteria, particularly when non-metabolized drug residues are present.” in revised version (page 3; Line 116-123).

Point 15: 125: what do authors mean with “insufficient extraction and quantification procedures”? recovery % is used to correct results in field samples, so that they can be compared (see also 130-131)

Response 15: Thank you for your constructive suggestion. The sentences have been added with more details “the accuracy and reliability of the reported amounts of antibiotic excreted by animals may be influenced by variations in the extraction and quantification procedures used in these investigations. It is important to note that suboptimal extraction and quantification methods can result in a wide range of observed values and may not accurately reflect the true levels of antibiotic excretion. Therefore, further optimization and standardization of these procedures is necessary to improve the precision and comparability of results across studies.” in revised version (page 3; Line 129-135).

Point 16: 147-151: sentence difficult to understand, please rephrase; furthermore, “macrolide-lincosamidreptogramin B” is repeated twice, “-lactams” needs a “beta”, and check lincosamidreptogramin”

Response 16: Thank you for pointing out the mistake. The one “macrolide-lincosamidreptogramin B” has been deleted and, “β-lactams” has been added in revised version (page 5; Line 155-157).

Point 17: 152-154: the classes of antibiotics used in the different species vary, what does the cumulative "average antimicrobial agent consumed per kilogram of animal produced" mean? and please, check for sentence construction

Response 17: Thank you for your constructive suggestion. The more details has been added “Van Boeckel et al.[1] conducted an analysis of antimicrobial consumption in animal production systems, and observed significant differences in average usage rates across species. Specifically, the mean amount of antimicrobial agents consumed per kilogram of animal produced was estimated to be 45 mg/kg for cattle, 148 mg/kg for chickens, and 172 mg/kg for pigs.” in revised version (page 5; Line 158-162).

Point 18: 157-158: sentence difficult to understand, please rephrase

Response 18: We are very thankful to the reviewer for highlighting this point. The sentence has been rephrased “Chickens are raised using antibiotics throughout their life cycle, therefore their feces may contain more ARG than other livestock feces after two months of hatching to slaughter.” in revised version (page 5; Line 165-167).

Point 19: 160-161: please, explain the sentences (and the meaning of “liver abscesses”, as the main bacterial disease of dairy caws is mastitis)

Response 19: Thank you for commenting on this point. We agree with your point of view regarding the main bacterial disease of dairy caws is mastitis. However, Mastitis is an inflammation of the mammary gland and udder tissue. It can be caused by a variety of factors including bacteria, viruses, and fungi. Liver abscesses are caused by bacteria that inhabit rumen lesions caused by acidosis and subsequently escape into the bloodstream. Liver abscesses are seen in all ages and breeds of cattle wherever cattle are raised. They are most common in feedlot and dairy cattle fed rations that predispose to rumenitis. The “Mastitis” has been added in the revised version (page 5; Line 170-171).

Point 20: 162-173: paragraph difficult to understand, please rephrase

Response 20: We are very thankful to the reviewer for highlighting this point. The sentences have been rephrased “ARGs varies widely across countries and regions, with some areas experiencing higher levels of ARGs than others. According to a recent meta-analysis of 96 countries, China was found to have the highest absolute abundance of ARGs, likely due to its status as the world's largest producer and consumer of antibiotics [31]. Despite similar levels of antibiotic consumption on a global scale (with both Shandong and Colorado accounting for 23% and 19% of worldwide use in 2010, respectively), there is no correlation between antibiotic consumption and the abundance of antibiotic resistance genes (ARGs). This is exemplified by the fact that Shandong's ARG abundance is ten times higher than that of Colorado, contradicting the notion that higher antibiotic consumption necessarily leads to increased ARG prevalence [32]. The abundance of antibiotic resistance genes (ARGs) can vary significantly between countries, influenced by country-specific factors. For instance, the quantity of ARGs present in cattle dung from Shandong is approximately 100 times greater than that found in Shaanxi (around 1011 copies/g) [33]. ARG abundance can vary significantly between dairy farms in different regions, as seen in the differences between Midwest and Northeastern farms. Although there may be some correlation between ARG abundance and antibiotic use or residual antibiotics in certain countries, the proliferation of ARBs and the dynamics of ARG propagation and attenuation are influenced by site-specific physical and chemical factors, leading to considerable variability.” in the revised version (page 5; Line 172-190).

Point 21: 176: what is the meaning of “essential pathogens”? please, explain

Response 21: Thank you for your comment. The sentence has been rephrased as “If pathogens are not effectively inactivated prior to the use of manure in crop cultivation or discharge into the environment, there is a risk of foodborne illness due to the potential contamination of crops or water sources. Common foodborne pathogens, such as Salmonella, E. coli, and Listeria, can be present in biowaste and proper handling and treatment of this waste is crucial to prevent the spread of pathogens and protect public health.” in the revised version (page 6; Line 193-198).

Point 22: 177-179: sentence difficult to understand, please rephrase

Response 22: Thank you for your comment. The sentence has been rephrased “Pathogens present in food, water, or air can pose a risk to human health when proper waste management and treatment practices are not followed. Factors such as the increase in food waste, animal manure, and other biowaste, combined with inadequate pathogen inactivation during waste management, can contribute to the spread of biosafety-related diseases.” in the revised version (page 6; Line 217-226).

Point 23: 190-198: paragraph difficult to understand, please rephrase

Response 23: The authors would like to thank the reviewer for providing his valuable suggestions. The sentence has been rephrased “To achieve effective pathogen inactivation while producing bio-methane, it is crucial to comprehend and measure the impact of different operational conditions on the inactivation of pathogens. Gurmessa et al.[34] performed a meta-analysis on various substrates and found a statistically significant impact of AD on pathogen inactivation (P < 0.01). The study included 63 investigations of fecal coliform inactivation, 51 investigations of E. coli inactivation, and 38 investigations of salmonella inactivation. More data was available for mesophilic AD processes than for either ambient or thermophilic AD processes, indicating a preference for mesophilic systems. Understanding and quantifying the effect of different parameters on pathogen inactivation is critical for balancing the operating conditions of pathogen inactivation and bio-methane production.” in the revised version (page 6; Line 235-244).

Point 24: 204-205: please explain “moderate, mesophilic ant thermophilic” conditions

Response 24: Thank you for your comment. In AD, Mesophilic conditions are those that promote the growth of microorganisms at temperatures between 25 - 40 °C. This range of temperature is the most common in AD and is ideal for the growth of a wide range of microorganisms. This temperature range is also optimal for the digestion of most organic materials.

Thermophilic conditions, on the other hand, promote the growth of microorganisms at temperatures between 45-65 °C Thermophilic conditions are more effective than mesophilic conditions in breaking down complex organic compounds such as proteins, lipids, and lignocellulose, and are ideal for treating certain types of waste, such as sewage sludge or food waste.

Moderate conditions are those that fall between mesophilic and thermophilic conditions, typically between 40-45 °C. The temperatures for moderate, mesophilic ant thermophilic have been added in the revised version (page 6; Line 250-251).

Point 25: 232-233: quite obvious, as microbiomas of different species are different

Response 25: Thank you for sharing that information with me. It’s true that “microbiomas of different species are different” more explanation has been added “The study conducted by Sun et al. [45] utilized meta-analysis to demonstrate that thermophilic anaerobic digestion (AD) is a more efficient means of reducing antibiotic resistance genes (ARGs) in pig manure when compared to cattle manure. This finding implies that the type of feedstock employed has a significant impact on determining the most suitable operating temperature for the effective removal of ARGs.” in the revised version (page 6; Line 277-281).

Point 26: Table 3: please, make caption more descriptive of table content

Response 26: Thank you for your comment. It has been changes in revised version “The effect of anaerobic digestion on antibiotic residues in manure: reduction after in concentrations anaerobic digestion. (Page 6; Line 359-360).

Point 26: 316: 3.2 section is more explanatory of the procedure description, so maybe it could be convenient to move it as 3.1, to improve the comprehension of the text

Response 26: Thank you for your suggestions. The section 3.2 section has been replaced with section 3.1.

Point 27: 353-354: this is a scientific paper, maybe the reader already has some knowledge about biotic and abiotic

Response 27: Thank you for your suggestions. The biotic and abiotic have been deleted in the revised version (page 6; Line 413-414).

Point 28: 361-363: sentence out of context

Response 28: Thank you for your suggestions. The sentence has been deleted in in the revised version.

Point 29: 379-388: paragraph difficult to understand, please rephrase

Response 29: Thank you for your suggestions. The paragraph “In conclusion, it was evident that optimizing various process parameters could successfully eliminate ARGs during AD. Previous research has identified certain fundamental factors that influence the proliferation of ARGs in AD. However, the interactions between different process factors often led to contradictions in prior findings. Therefore, further research is required to fully comprehend how the interplay between various process factors impacts the proliferation of ARGs in AD. Additionally, continued development of conductive additives and pre-treatment procedures is crucial to provide dual solutions for better biogas generation and ARG management, ensuring economic sustainability of AD while also decreasing the danger of digestate for land application.” has been rephrased in in the revised version (page 6; Line 440-448).

Reviewer 2 Report

This paper, entitled The antibiotics degradation and its mechanisms during the livestock manure anaerobic digestion, is a scholarly work and can increase knowledge on this domain. The authors provide an interesting and original study, the content is relevant to Molecules.

The manuscript is quite well written and well related to existing literature. The abstract and keywords are meaningful.

I have some general and specific comments:

- Why not including antibiotics used in poultry? The authors mentioned that they focused on antibiotics used in pigs and cattles.

- Please give some precision about the legend used in Table 1 and please provide references for each antibiotic listed in this Table 1.

- Is this paper based only on literature or also on experimental works carried out by the authors? If this work is based on literature, from my point of view, it could be interesting to provide also scientometric or bibliometric data for a specific period of time (occurence of terms, number of papers per year, ...). What were the keywords used for the search of papers? Is there specific database used or not?

- Is this work focused on a specific location or area?

- Is there any step of hygienization applied before AD process for some of papers found dealing with this topic? If yes, what was the effect of such thermal treatment on antibiotics?*

- What about the effect of the temperature applied during the AD process (mesophilic vs thermophilic conditions) if this information was given in papers?

- What are the levels of antibiotics in digestate? is there some antibiotics still remaining in digestate? what about risk of concentration or accumulation?

- What about the degraded molecules obtained after antibiotics degradation?

- Is there some specific methods for the destruction of antibiotics (different of biodegradation)? ozonation, chemical, thermal treatments?

- What about the regulation concerning the presence or the concentration of antibiotics after AD process in digestate, especially if there's a spreading of digestate on fields? Is there any regulation or specifications?

As it, this paper is not fully acceptable for publication and requires some amendments and additional information. I recommend the following decision: RECONSIDER AFTER MAJOR REVISION.

Author Response

Response to Reviewer 2 Comments

This paper, entitled The antibiotics degradation and its mechanisms during the livestock manure anaerobic digestion, is a scholarly work and can increase knowledge on this domain. The authors provide an interesting and original study, the content is relevant to Molecules.

The manuscript is quite well written and well related to existing literature. The abstract and keywords are meaningful.

Response: Thank you for your positive feedback on our paper entitled "The antibiotics degradation and its mechanisms during the livestock manure anaerobic digestion." We are glad to hear that you found our study interesting and original, and that the content is relevant to Molecules. It is appreciated that you found the manuscript well written, well-referenced, and well-referenced to existing literature as well as overall quality. In order to convey the main focus of our study, we worked hard to ensure that the abstract and keywords were meaningful. Our sincere thanks go out to you for reviewing our paper and for your valuable feedback.

Point 1: Why not including antibiotics used in poultry? The authors mentioned that they focused on antibiotics used in pigs and cattle.

Response 1: Thank you for your question. The paper reports on the degradation of antibiotic resistance genes (ARGs) during anaerobic digestion. The limited availability of data on poultry manure anaerobic digestion compared to swine and cattle manure may be attributed to the high nitrogen content in poultry manure, which makes it more suitable for composting than anaerobic digestion.

Point 2: Please give some precision about the legend used in Table 1 and please provide references for each antibiotic listed in this Table 1.

Response 2: Thank you for pointing out this lapse. The legend has been added in the table and reference is also added in revised version.

Point 3: Is this paper based only on literature or also on experimental works carried out by the authors? If this work is based on literature, from my point of view, it could be interesting to provide also scientometric or bibliometric data for a specific period of time (occurence of terms, number of papers per year, ...). What were the keywords used for the search of papers? Is there specific database used or not?

Response 3: We are very thankful to the reviewer for highlighting this point. Yes, we tried our best to cover all data available on this topic. It is important to understand that this work is designed on the degradation of ARGs from manure under AD condition, which is relatively less reported in the previous studies as compared to composting of manure. The data we presented here specifically cover each aspect of the aforementioned topic in details. The literature used to cover several aspects of this review was obtained using the following search terms, indexes searched, number of papers found and used.

  1. Search terms: Anaerobic digestion, manure, antibiotic resistance genes (ARGs), degradation, antibiotics, livestock waste, veterinary
  2. Indexes search: Google Scholar, Scopus, PubMed, DOAJ, ISI Indexing, SCI,and  SCIE.
  3. A total number of relevant literatures used and found was approximately 40 for this work.

Point 4: Is this work focused on a specific location or area?

Response 4: Thank you for question. The scope of this review paper is to investigate the degradation dynamics of antibiotic resistance genes (ARGs) during anaerobic digestion of livestock waste. Notably, this paper does not encompass the examination of composting processes. Despite a wealth of studies on the effects of composting on ARGs, limited attention has been given to the fate of ARGs during anaerobic digestion, leading to a paucity of data on this subject.

Point 5: - Is there any step of hygienization applied before AD process for some of papers found dealing with this topic? If yes, what was the effect of such thermal treatment on antibiotics?

Response 5: Thank you for questions. I found a review paper that specifically discusses the hygienization of biowastes for anaerobic digestion. Anaerobic digestion of biowastes using hygienization-related technologies is discussed in this paper. Moreover, the paper examined global regulation and the energy consumption of thermal pasteurization in biogas plants.  There are no specific papers that address the effects of thermal treatment on antibiotics during livestock waste hygienization before anaerobic digestion.

Point 6: What about the effect of the temperature applied during the AD process (mesophilic vs thermophilic conditions) if this information was given in papers?

Response 6: Thank you for your inquiry. This paper extensively examines the impacts of mesophilic and thermophilic conditions, as well as moderate conditions, on the degradation of antibiotic resistance genes (ARGs), providing detailed insights on these factors and their influence on ARG degradation.

Point 7:  What are the levels of antibiotics in digestate? is there some antibiotics still remaining in digestate? what about risk of concentration or accumulation?

Response 7: Thank you for your inquiry. There are several factors that can affect the level of antibiotics in digestate, the end product of anaerobic digestion, including the type and quantity of antibiotics used in the animal feed, the efficiency of digestion, and the duration of storage after digestion. Although digestate is usually lower in concentration than original manure, some studies report detectable levels of antibiotics and their metabolites. Farm or facility management practices may impact the concentration or accumulation of antibiotics in digestate. The soil and surrounding environment can become contaminated with antibiotics and their metabolites if digestate is not properly treated or managed, such as being land-applied in excess amounts or without sufficient time for degradation. In addition, antibiotic resistance in microorganisms may develop as a result. A deeper understanding of antibiotic levels and fate in digestate, as well as their potential environmental impacts, requires further research.

Point 8: What about the degraded molecules obtained after antibiotics degradation?

Response 8: Thank you for your inquiry. The degradation of antibiotics can produce degradation products that may still have antibiotic activity. These degradation products may also be more mobile and persistent in the environment than the parent antibiotics.

Point 9: Is there some specific methods for the destruction of antibiotics (different of biodegradation)? ozonation, chemical, thermal treatments?

Response 9: Thank you for your inquiry. While several methods exist for the destruction of antibiotics that are distinct from biodegradation, such as ozonation, chemical treatments (e.g., hydrogen peroxide, chlorine dioxide, and potassium permanganate), thermal treatments (e.g., incineration and autoclaving), and photodegradation, our paper primarily focuses on the degradation of ARGs through anaerobic digestion. ARGs are genetic elements that persist in the environment even after antibiotic compounds have been degraded. Thus, the effective degradation of ARGs may require specialized methods, such as advanced oxidation processes and microbial technologies like bioaugmentation or biostimulation.

Point 10: What about the regulation concerning the presence or the concentration of antibiotics after AD process in digestate, especially if there's a spreading of digestate on fields? Is there any regulation or specifications?

Response 10: Thank you for your inquiry. Regulations and guidelines exist pertaining to the levels and prevalence of antibiotics in digestate subsequent to anaerobic digestion (AD) and its subsequent application on agricultural land. National legislation or guidelines regulate the utilization of digestate as a fertilizer in numerous countries. The European Fertilizer Regulation (EC) No. 2003/2003 governs the utilization of digestate as a fertilizer in the European Union. This regulation establishes thresholds for the occurrence of impurities, such as antibiotics, in fertilizers. The regulation of biosolids, including digestate, in the United States falls under the purview of the Environmental Protection Agency (EPA) through the Clean Water Act and the National Pollutant Discharge Elimination System (NPDES) permit program. The Environmental Protection Agency (EPA) establishes regulatory thresholds governing the concentration of contaminants, such as antibiotics, in biosolids, in order to guarantee their safety for employment as a soil amendment. Furthermore, certain nations have established particular regulations or guidelines pertaining to the existence of antibiotics within digestate. The Federal Ministry for the Environment, Nature Conservation, and Nuclear Safety (BMU) in Germany has released guidelines pertaining to the utilization of digestate as a fertilizer. These guidelines incorporate restrictions on the existence of antibiotics.

Reviewer 3 Report

Find the specific comment to the authors for more details;

1.   1. The Abstract really needs revision and writes more interestingly by providing major findings.

2.     2. Page 3 Line 109-113 “In Shandong, China, 126 pig manure samples taken from 21 … unaltered “or or” as living byproducts of the parent species. [23]. Please correct it.

3.     3. Revise the Discussion and write more interestingly. I would suggest citing recent papers to improve the discussion.

4.      The degradation of ARGs during anaerobic digestion and the potential impact on microbial community or ARGs have been intensively investigated with various focuses, for instance temp, Operating temperature, Two-stage anaerobic digestion, Residence time, and Total Solids.  The authors should define the significance of the study also needs to be discussed eg., Understanding their fate during anaerobic digestion has become significant research. 

Author Response

Response to Reviewer 3 Comments

Point 1: The Abstract really needs revision and writes more interestingly by providing major findings.

Response 1: Thank you for your question. The paper reports on the degradation of antibiotic resistance genes (ARGs) during anaerobic digestion. The limited availability of data on poultry manure anaerobic digestion compared to swine and cattle manure may be attributed to the high nitrogen content in poultry manure, which makes it more suitable for composting than anaerobic digestion.

Point 2: Page 3 Line 109-113 “In Shandong, China, 126 pig manure samples taken from 21 … unaltered “or or” as living byproducts of the parent species. [23]. Please correct it.

Response 2: Thank you for pointing out this lapse. The one “or” has been deleted in the table and in revised version.

Point 3: Revise the Discussion and write more interestingly. I would suggest citing recent papers to improve the discussion.

Response 3: Thank you for your suggestions. The paragraph “In conclusion, it was evident that optimizing various process parameters could successfully eliminate ARGs during AD. Previous research has identified certain fundamental factors that influence the proliferation of ARGs in AD. However, the interactions between different process factors often led to contradictions in prior findings. Therefore, further research is required to fully comprehend how the interplay between various process factors impacts the proliferation of ARGs in AD. Additionally, continued development of conductive additives and pre-treatment procedures is crucial to provide dual solutions for better biogas generation and ARG management, ensuring economic sustainability of AD while also decreasing the danger of digestate for land application.” has been rephrased in in the revised version (page 6; Line 440-448).

Point 4: The degradation of ARGs during anaerobic digestion and the potential impact on microbial community or ARGs have been intensively investigated with various focuses, for instance temp, Operating temperature, Two-stage anaerobic digestion, Residence time, and Total Solids.  The authors should define the significance of the study also needs to be discussed eg., Understanding their fate during anaerobic digestion has become significant research. 

Response 4: Thank you for your question. The paper reports on the degradation of antibiotic resistance genes (ARGs) during anaerobic digestion. The limited availability of data on poultry manure anaerobic digestion compared to swine and cattle manure may be attributed to the high nitrogen content in poultry manure, which makes it more suitable for composting than anaerobic digestion.

Round 2

Reviewer 1 Report

Thanks to the authors for accepting the observations, the fluency and comprehensibility of the text gains a lot. Only very few recommendations remain, after which the manuscript can be accepted. No need for further review. 

58: again, the term “sub-lethal” is misleading, as it seems to refer to the animals themselves, in which the administered dose must be very far from the lethal one; if the authors are referring to bacteria, it would be more appropriate to call it “sub-therapeutic”

156: please, check the spelling for macrolide-lincosamide-streptogramin B

156: are “(fluoroquinolone, chloramphenicol, and amphenicol) the subject of FCA ARG of 154?

250-251: explanation provided by the authors in Response 24 is very explicative, perhaps it should also be available to the reader, not just the reviewer, so it is recommended to add it to the text

Author Response

Thanks to the authors for accepting the observations, the fluency and comprehensibility of the text gains a lot. Only very few recommendations remain, after which the manuscript can be accepted. No need for further review. 

Response: Thank you for your thoughtful review of our manuscript. We appreciate your kind words and are pleased to hear that the revisions we made based on your feedback have improved the fluency and comprehensibility of the text. We are grateful for the opportunity to receive your additional recommendations to enhance the manuscript's quality further. We will carefully consider your feedback and make the necessary revisions to ensure that the manuscript meets the highest standards of scientific rigor. We are pleased to hear that you do not recommend a further review and that the manuscript is close to being ready for acceptance. We will ensure that we address the remaining recommendations promptly and thoroughly and work diligently to make the necessary revisions. Once again, we appreciate your constructive feedback and valuable guidance in improving our manuscript. We look forward to submitting the revised manuscript for your final evaluation.

Top of Form

Bottom of Form

Point 1: 58: again, the term “sub-lethal” is misleading, as it seems to refer to the animals themselves, in which the administered dose must be very far from the lethal one; if the authors are referring to bacteria, it would be more appropriate to call it “sub-therapeutic”

Response 1: Thank you for bringing this to our attention. We appreciate your feedback and have made the necessary correction in the revised version of the manuscript. Specifically, we have replaced the word "sub-lethal" with "sub-therapeutic," as recommended (see page 1, line 58).

Point 2:156: please, check the spelling for macrolide-lincosamide-streptogramin B

Response 2: We are thankful for your careful review and for bringing this error to our attention. We apologize for any confusion arising from our error and appreciate the opportunity to correct it. Specifically, we have replaced the incorrect term "macrolide-lincosamidrepto-gramin B" with the correct term "macrolide-lincosamide-streptogramin B" in the revised version of the manuscript. (See page 5, line 159).

Point 3:156: are “(fluoroquinolone, chloramphenicol, and amphenicol) the subject of FCA ARG of 154?

Response 3: Thank you for your insightful observation and for bringing this to our attention. We agree that the FCA measurements were the same and were inadvertently repeated in the manuscript. We have made the necessary correction in the revised version of the manuscript (See page 5, line 159).

Point 4: 250-251: explanation provided by the authors in Response 24 is very explicative, perhaps it should also be available to the reader, not just the reviewer, so it is recommended to add it to the text

Response 4: Thank you for your feedback on Response 24. We appreciate your suggestion and agree that the explanation may be useful to readers beyond the reviewer. In response to your recommendation, we have included the additional explanation “In AD, Mesophilic conditions are those that promote the growth of microorganisms at temperatures between 25 - 40 °C. This range of temperature is the most common in AD and is ideal for the growth of a wide range of microorganisms. This temperature range is also optimal for the digestion of most organic materials. Thermophilic conditions, on the other hand, promote the growth of microorganisms at temperatures between 45-65 °C Thermophilic conditions are more effective than mesophilic conditions in breaking down complex organic compounds such as proteins, lipids, and lignocellulose, and are ideal for treating certain types of waste, such as sewage sludge or food waste. Moderate conditions are those that fall between mesophilic and thermophilic conditions, typically between 40-45 °C.” in the revised version of the manuscript, to make it available to all readers (See page 7, lines 257-266).

Reviewer 2 Report

The authors provide a revised version of their manuscript taking into account all the comments and requests of amendments made in the previous review. The authors provide also detailed and justified answers to all the comments and requests of amendments, I agree with all these answers. As it, the paper is now fully acceptable for publication and I recommend the following decision: ACCEPT IN PRESENT FORM.

Author Response

The authors provide a revised version of their manuscript taking into account all the comments and requests of amendments made in the previous review. The authors provide also detailed and justified answers to all the comments and requests of amendments, I agree with all these answers. As it, the paper is now fully acceptable for publication and I recommend the following decision: ACCEPT IN PRESENT FORM.

Response: We would like to express our sincere gratitude for your insightful comments and constructive feedback on our manuscript. We appreciate the time and effort you have invested in reviewing our work, and we are pleased to hear that the revised version has adequately addressed all of your concerns. We have carefully considered each of your comments and requests for amendments and have made the necessary revisions to improve the manuscript's quality. We have also provided detailed and justified responses to all of your comments and requests, and we are delighted to hear that you found them satisfactory. Your positive assessment that the paper is now fully acceptable for publication is gratifying, and we are thrilled to hear that you recommend the paper's acceptance in its present form. We value your expert opinion, and we are grateful for your guidance in making this manuscript ready for publication. Once again, we thank you for your valuable feedback and look forward to seeing our manuscript published soon.
